# State-recycling and time-resolved imaging in topological photonic lattices

Sebabrata Mukherjee [1], Harikumar K. Chandrasekharan [1], Patrik Öhberg[1],

Nathan Goldman[2] & Robert R. Thomson[1]

Photonic lattices—arrays of optical waveguides—are powerful platforms for simulating a range of phenomena, including topological phases. While probing dynamics is possible in these systems, by reinterpreting the propagation direction as time, accessing long timescales constitutes a severe experimental challenge. Here, we overcome this limitation by placing the photonic lattice in a cavity, which allows the optical state to evolve through the lattice multiple times. The accompanying detection method, which exploits a multi-pixel single-photon detector array, offers quasi-real time-resolved measurements after each round trip. We apply the state-recycling scheme to intriguing photonic lattices emulating Dirac fermions and Floquet topological phases. We also realise a synthetic pulsed electric field, which can be used to drive transport within photonic lattices. This work opens an exciting route towards the detection of long timescale effects in engineered photonic lattices and the realisation of hybrid analogue-digital simulators.

[1] Scottish Universities Physics Alliance (SUPA), Institute of Photonics and Quantum Sciences (IPaQS), School of Engineering & Physical Sciences, Heriot-Watt University, Edinburgh EH14 4AS, UK. [2] Center for Nonlinear Phenomena and Complex Systems, Université Libre de Bruxelles, CP 231, Campus Plaine, 1050 Brussels, Belgium. Correspondence and requests for materials should be addressed to S.M. (email: mukherjeesebabrata@gmail.com) or to N.G. (email: ngoldman@ulb.ac.be) or to R.R.T. (email: r.r.thomson@hw.ac.uk)

In the last decade, topological photonics has emerged as a promising field for the realisation and detection of exotic states of matter with topological properties[1-3]. Building lattices for light has in particular allowed for the engineering of topological phases that have remained inaccessible in solid-state devices, such as Floquet topological phases[4-7], and has offered novel methods by which the geometry and topology of Bloch bands can be directly extracted[8-10]. Among the various photonic devices developed so far, photonic lattices, consisting of periodic arrays of coupled optical waveguides, provide a particularly rich toolbox for the simulation of intriguing toy-models of topological phenomena[5-7]. Often realised using ultrafast-laser-fabrication techniques[11], these engineered lattices allow for independent and dynamical control over the effective inter-site tunnelling and on-site potentials, and can be arranged into various geometries. Beyond topological effects, photonic lattices have also been exploited to investigate many other effects[12,13] including quantum correlations[14,15] and the photonic Zeno effect[16].

In the scalar-paraxial approximation, light propagation across a photonic lattice is governed by a Schrödinger-like equation[13], where the propagation distance ($z$) plays the role of time ($z \leftrightarrow t$). In current photonic lattice simulators, unlike fibre networks[17], the effective time-evolution of a specific input state is measured over relatively short timescales, which are set by the maximum propagation distance $L \approx 10$ cm of the fabricated lattices. This approach complicates, or even prevents, the observation of physical phenomena that are associated with slow dynamics, such as those emanating from weak effective inter-particle interactions[18,19] or weakly dispersive bands[20]. In addition, it prevents the study of topological edge modes over long durations, and in particular topological interference effects[21].

Here, we propose to overcome this limitation by placing the photonic lattice in an optical cavity and recycling the optical state through the lattice multiple times. After each cavity round-trip, the time-evolved output state is then observed using pulsed excitation light and an advanced single-photon avalanche detector (SPAD) array[22], which facilitates independent time-correlated single-photon counting for each mode of the photonic lattice. We demonstrate the operation of two types of cavities, and apply these to study the quasi-real-time evolution of pseudo-relativistic modes and Floquet anomalous topological edge modes over long distances. We also show how synthetic electric fields can be naturally introduced in this platform, hence offering a simple method by which transport can be driven within photonic lattices. In principle, the output state could be finely modified after each round-trip, offering the possibility of engineering quantum walks, local dissipation, gauge fields and effective interaction effects in a (quasi-real-time) stroboscopic manner.

## Results
**State-recycling schemes**. The key concept behind our state-recycling system is to place the photonic simulator inside an optical cavity, which allows the output state to be fed back into the simulator. As shown in Fig. 1a, we consider two types of cavities. In the linear cavity scheme, light is simply reflected at both ends of the lattice, so that the effective time-evolution is dictated by an alternating sequence of time-evolution operators, $\hat{U}(t) = e^{-i\hat{H}_2 T/2} e^{-i\hat{H}_1 T/2} \dots e^{-i\hat{H}_2 T/2} e^{-i\hat{H}_1 T/2}$, where the Hamiltonians $\hat{H}_{1,2}$ are related by a time-reversal operation and $T$ is the period associated with each round trip. This linear cavity is most suitable to study the long-time dynamics associated with a specific engineered Hamiltonian $\hat{H}$, whenever the latter is time-reversal symmetric, $\hat{H} = \hat{H}_1 = \hat{H}_2$. In the ring cavity scheme, the output state is recycled and re-injected directly into the input [Fig. 1a]. This ring scheme is thus suitable to simulate

Hamiltonians without time-reversal-symmetry, such as those associated with the quantum Hall effect and (Floquet) Chern insulators[1,2]. By launching optical pulse trains at the input of a photonic lattice, we are able to use an advanced single-photon sensitive detector array to perform independent time-correlated single-photon counting for each mode of the simulator, and thus to observe the time-evolution of the light field in a quasi-real-time manner. The key technology at the heart of our scheme is therefore the single-photon sensitive detector array itself, which in our case consists of a $32 \times 32$ square array of silicon based SPADs manufactured using complementary metal oxide semiconductor (CMOS) technology. Each SPAD has a $\approx 6\,\mu m$ diameter photosensitive area and the pixel pitch is $50\,\mu m$. The photon detection efficiency of the SPADs is maximum at a wavelength of about 500 nm, but is still single-photon sensitive up to about 1000 nm. Each individual pixel can acquire time information with a resolution of 53 ps for 10 bits (i.e. 54 ns) temporal range. This type of detector array was recently used for a variety of multiplexed single-photon counting[23] applications, including light-in-flight imaging[24] and multiplexed single-mode single-photon-sensitive wavelength-to-time mapping[25].

**1D Dirac fermions**. We first apply the linear cavity scheme to a periodically driven lattice which emulates pure one-dimensional (1D) Dirac fermions[26-28]. In this model, the amplitudes of nearest-neighbour couplings are staggered and modulated in a periodic manner according to a two-step sequence [Fig. 1b]: for the first half a period ($0 \leq t \leq T/2$), neighbouring couplings are $J_1 = 0$ and $J_2 = \pi/T$, while for the remaining half ($T/2 \leq t \leq T$), these couplings are $J_1 = \pi/T$ and $J_2 = 0$. In this configuration, the effective Hamiltonian associated with stroboscopic motion[29] takes the form of a Dirac Hamiltonian, $\hat{H}_{\text{eff}} = v_D k \hat{\sigma}_z$, where $k$ is the crystal momentum, $\hat{\sigma}_z$ is a Pauli matrix describing the lattice pseudo-spin, with the effective speed of light $v_D = 2d/T$ where $d$ is the lattice spacing (see Supplementary Note 1). Accordingly, the Floquet spectrum consists of two linearly dispersive bands, Fig. 1d, indicating that a single-band excitation is expected to travel along the lattice without any diffraction.

To implement this driving protocol, a photonic lattice of 24 sites was fabricated inside a 15-mm-long borosilicate substrate using ultrafast laser inscription[11] (see Methods). Each waveguide pair was synchronously curved to spatially and dynamically turn on/off any particular bond [Fig. 1c], hence generating the desired effective couplings[6]. Both facets of the substrate were polished and silver-coated (with $\approx 90\%$ reflectivity) so as to form the above-mentioned linear cavity, and the optical mode of each waveguide was imaged from the output of the lattice onto individual SPADs of the Megaframe (MF32)[22], see Supplementary Fig. 1. In our setup, the actual photonic lattice only describes half of the complete driving sequence described above (i.e. $L \equiv T/2$). However, thanks to the linear cavity, and due to the time-reversal nature of the underlying effective Hamiltonian, our photonic lattice enables us to launch an initial state at the effective time $t = 0$ and to detect it at stroboscopic times $t = (1/2 + N)T$, where $N$ is a positive integer.

In the experiment, we launched $873 \pm 3$ nm pulses at the edge of the lattice, and we measured the time-correlated photon counts up to four and a half round trips. Effectively, this corresponds to exploring nine times the physical length of the lattice (i.e. $9L = 135$ mm), see Fig. 1e–i. As shown, the data recorded by individual pixels provide spatial as well as temporal intensity distributions, see Supplementary Fig. 2. The number of round trips that can be observed in our current experimental system depends on the quality factor of the cavity, which is primarily determined by the waveguide losses ($\approx 5$ dB per each round trip for this experiment).

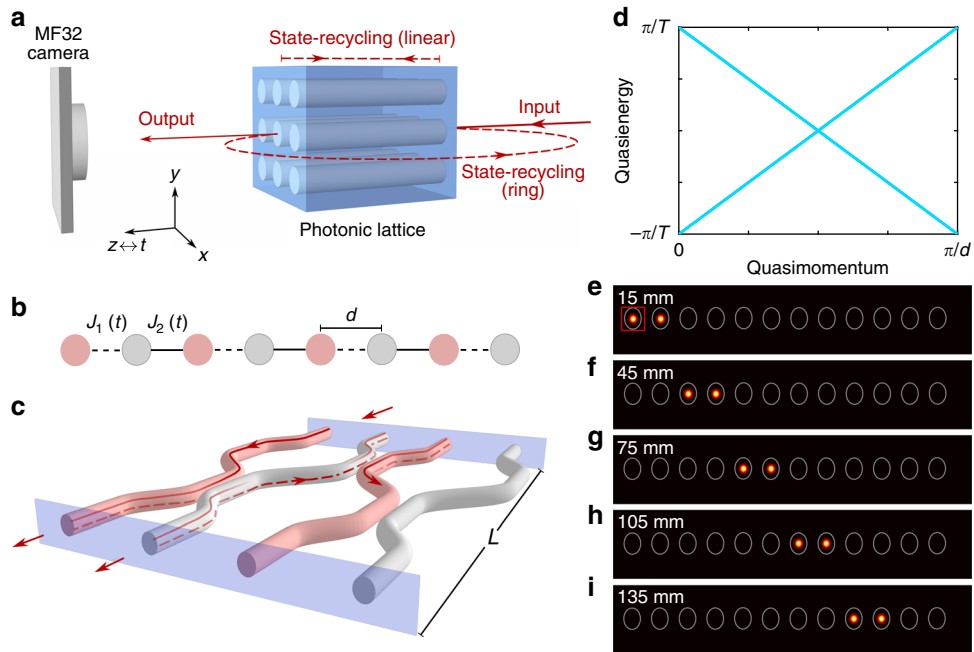

**Fig. 1** State-recycling techniques. **a** Simplified sketch illustrating the experimental technique to detect time evolution in a photonic lattice through state-recycling. Both the linear and ring recycling schemes are illustrated. The propagation distance is the analogous time ($z \leftrightarrow t$). **b** One-dimensional driven lattice with nearest-neighbour couplings $J_{1,2}$, which are varying periodically in time. **c** Photonic implementation of the driven lattice in **b**. Here, the state-recycling is performed using a linear cavity, the facets of which are indicated by two parallel blue planes. **d** Floquet spectrum of the driven lattice in **b**, **c** consists of two linearly dispersive bands for the following driving protocol: $J_{1,2} = 0$, $\pi/T$ for $0 \leq t \leq T/2$ and $J_{1,2} = \pi/T$, 0 for $T/2 \leq t \leq T$ where $T$ is the driving period. **e–i** Excitation of a linearly dispersive band. Experimentally observed intensity distributions at $t = (1/2 + N)T$, $N = 0, 1, 2, 3$. The red square indicates the waveguide which was excited at the input. The effective propagation distances are indicated on each image

In other words, the temporal data can be accessed until the detected photon count is comparable to the noise level. Exploiting suitable detectors and performing the experiment at a longer wavelength (e.g. near 1550 nm, where comparatively low-loss waveguides can be fabricated) will allow one to access many more round trips.

**Discrete diffraction and synthetic electric fields**. A similar linear cavity setup was used to investigate discrete diffraction in a 1D photonic lattice consisting of twenty coupled single-mode waveguides with 19 μm waveguide-to-waveguide spacing and length $L = 30$ mm. In this situation, the propagation of optical fields emulates the motion of a particle in a (single-band) tight-binding lattice[12] with an analogous tunnelling strength $J = 0.029$ mm$^{-1}$. Figure 2a–d show the intensity patterns that were experimentally obtained for four effective propagation distances, $L = 30$ mm, $3L = 90$ mm, $5L = 150$ mm and $7L = 210$ mm. Interestingly, as shown in Fig. 2e–h, these intensity patterns (red bars) start to deviate from the naively expected distributions (cyan) for effective propagation distances $z > 3L$. This is due to the small angles located at the input and output facets ($\approx \pm 0.1°$, respectively) of the substrate (see inset of Fig. 2e), which cause a linear variation of the optical phase along the lattice axis. These angles effectively produce a time-periodic (pulsed) electric field that acts on the particle along the lattice axis; see Supplementary Notes 3 and 4 and Supplementary Figs. 5–7 for more details. This picture is validated in Fig. 2e–h, which indicates that the experimental intensity patterns (red bars) agree well with our numerical simulations upon adding the effects of the pulsed electric field (blue bars). The resulting motion is found to correspond to approximately half the period of a Bloch oscillation; see Supplementary Note 4. We note that effects of the small facet angles

were not detected in the previous experiment (Fig. 1e–i) because the spatial extent of the analogous wavefunction was significantly smaller as compared to the current experiment (Fig. 2).

**Anomalous Floquet topological insulator**. Next, we demonstrate the operation of the ring cavity state-recycling method. First of all, using a symmetric directional coupler, we verified that both the phase and intensity of an optical state are recycled in the ring cavity as required. The electric field envelope at the two waveguides of the coupler is governed by the following coupled mode equations[12]: $i\partial_{z \leftrightarrow t}\psi_{1,2} = -J\psi_{2,1}$. By placing the coupler inside the ring cavity and launching light into one of the waveguides $\psi_1(z = 0) = 1$, we measure, in a time-resolved manner, the output intensities $|\psi_{1,2}|^2$ at both waveguides as well as the interference pattern between the waveguide modes in the far field i.e. the relative phase between $\psi_{1,2}$; see Methods and Supplementary Fig. 3 for more details. The agreement between theory and experimental results demonstrate that the evolution operator describing the re-injection of the state from the output of the coupler to the input preserves the relative phase and intensity after renormalisation accounting for the state-recycling losses.

Next, we focus on the application of the ring-type cavity to a two-dimensional periodically driven system exhibiting non-trivial topology[30,31]. The model realised in our photonic setup was introduced in ref.[31] to demonstrate the existence of anomalous topological edge modes, which are topological states appearing in periodically driven (Floquet) systems with no static-system counterparts. Formally, these robust propagating states are protected by a topological winding number, which, in contrast to the more conventional Chern number[32], takes into account the full-time dynamics of the time-modulated system[31]. Such anomalous topological edge modes were experimentally

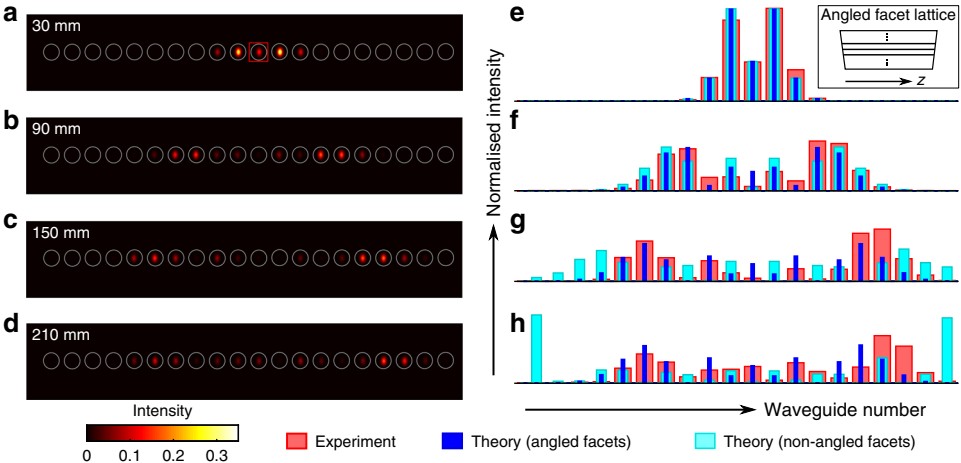

**Fig. 2** Discrete diffraction in the presence of a synthetic electric field. **a–d** Quasi-real-time evolution of light intensity in a 1D straight photonic lattice consisting of twenty coupled single-mode waveguides. The small facet angles of the substrate (inset in **e**) cause a linear phase shift along the lattice at each facet, which effectively produces a time-periodic (pulsed) synthetic electric field. **a–d** Show the output intensity distributions that were experimentally obtained after different effective propagation distances indicated on each image. Light was launched at a single waveguide indicated by the red square. **e–h** Comparison between the experimental observations in **a–d** and the associated numerical results. The vertical axis is the normalised optical intensity and the horizontal axis is the waveguide number. The measured intensity patterns (red bars) agree with the numerical simulations (blue) upon adding the effects of a pulsed (synthetic) electric field. The dynamics corresponds to approximately half a period of a Bloch oscillation, which is produced by the synthetic electric field. For comparison, we show the numerical results when omitting the pulsed electric field (cyan bars), where the Bloch oscillation is absent

demonstrated in photonics[6,7,33]. Here, we describe how our state-recycling technique can be applied to such intriguing states of matter, and exploit to reveal the quasi-real-time imaging of the corresponding chiral topological edge modes. We point out that the simulated system explicitly breaks time-reversal symmetry, and therefore it cannot be explored using the simpler linear cavity scheme.

The Floquet model of refs. [6,7,31] consists of a driven square lattice with four distinct nearest-neighbour couplings ($J_{1-4}$), which are varied in a circulating and time-periodic manner over the entire lattice, see Supplementary Note 2. When the driving period ($T$) is split into four equal steps, and upon the resonance condition $J_{1-4} = 2\pi/T$, chiral propagating anomalous edge modes are found to coexist with a perfectly localised bulk [see Supplementary Fig. 4a]. However, in this case, the Floquet bulk bands are degenerate at zero quasienergy, and an arbitrarily small deviation in the values of the parameters can potentially drive the system out from the anomalous regime[6]. To avoid possible ambiguity, and also for the sake of experimental practicability, we designed a slightly different model with $J_1 = 0$ and $J_{2,3,4} = 2\pi/T$. In this situation, maximally gapped bulk bands appear with zero Chern number [see Supplementary Fig. 4b], while the winding numbers associated with both the energy gaps (centred on 0 and $\pi/T$) are non-trivial and equal to one.

Such a photonic lattice of 63 sites [see Fig. 3a] was fabricated, corresponding to only two driving periods (i.e. $2T \leftrightarrow L = 70$ mm). Initially, all the waveguides are uncoupled at the wavelength range of interest [nearest-neighbour spacing is 40 μm]. Similar to the previous experiment (Fig. 1c), each waveguide pair is synchronously curved to spatially and/or dynamically turn on and off any particular bond[6]. To investigate the effect of a defect, one waveguide on the top-right edge [(8, 4) lattice site] was not fabricated. For the proposed experimental parameters, the edge modes, as well as the bulk states, can be efficiently excited by launching light into one optical waveguide.

In the experiment, we launch $780 \pm 5$ nm pulses of light at a desired site of the photonic lattice, which is placed inside the

ring-type cavity, see Supplementary Fig. 1. The output facet of the lattice is imaged onto the input of the lattice with unit magnification. For precise imaging, the input facet of the lattice is imaged on a CCD camera to observe the lattice sites, input state, and the output state after the first pass. Similar to the previous experiments, the optical mode of each waveguide from the output of the lattice is imaged onto the individual SPADs.

The edge modes are excited with ≈85% efficiency by launching light at the (4, 1) site on the bottom-right edge. Quasi-real-time chiral propagation of the edge modes is presented in Fig. 3b–e for four consecutive round trips (i.e. at analogous time $2T$, $4T$, $6T$ and $8T$). It should be noted that the group velocity of the edge modes along the top-right edge is twice that along the bottom-right edge[6]. For the proposed parameters, the non-dispersive bulk bands can be excited equally by launching light at a waveguide in the bulk [e.g. site (4, 5) in Fig. 3f, h]. In this situation, the input state is expected to cycle back to its initial position after two complete driving periods. Although the bulk bands are expected to be dispersionless for the proposed parameters, a small deviation from the desired parameters value makes both bands weakly dispersive[6] without altering the topological characteristics of the system. The effect of this weak dispersion of the bulk bands is hard to detect after first round trip, see Fig. 3f. However, the weak delocalisation of the bulk state becomes evident when exploiting the many round trips offered by the cavity [Fig. 3g, h], which demonstrates the capability of our state-recycling technique to detect the slow dynamics of such weakly dispersive states.

## Discussion

In conclusion, we have proposed and experimentally demonstrated a state-recycling technique based on time-correlated single-photon imaging, which enables us to measure the long-time dynamics of an input optical state propagating in an engineered photonic lattice. Importantly, this method introduces the possibility of detecting effective dynamics in a quasi-real-time (stroboscopic) manner. It offers a novel dimension to photonic

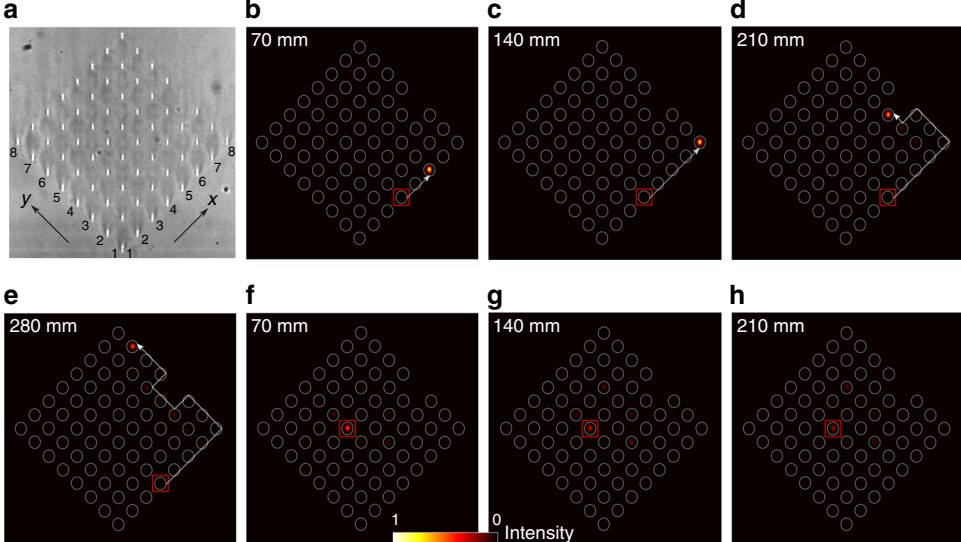

**Fig. 3** Quasi-real-time propagation of topological edge modes. **a** White-light micrograph of the facet of the driven square lattice. **b–e** One-way (here, counter-clockwise) propagation of the edge modes for effective times, $t = 2T$, $4T$, $6T$ and $8T$; here $2T \leftrightarrow L = 70$ mm. The edge modes are excited with ≈85% efficiency by exciting the (4, 1) site on the bottom-right edge (indicated by the red square). The edge modes are neither back-scattered by a corner nor by the defect [here, a missing waveguide at the (8, 4) site]. **f–h** Time evolution of the bulk state after times, $t = 2T$, $4T$ and $6T$. The weakly dispersive bulk bands are equally excited by launching light at the (4, 5) site. The delocalisation of the state becomes evident after long detection times

lattices, for which the final detection time was until now set by the length of the photonic lattice. Furthermore, the ring cavity method, in which the state is re-injected into the lattice in a controllable manner, offers a unique opportunity to design feedback mechanisms, i.e. a hybrid analogue-digital simulator. For instance, modifying the state after each round trip, according to some well-defined unitary operators, could be used as a simple protocol to design quantum walks[4,10], or could be suitably combined with another Floquet-engineering protocol. As we discussed, such stroboscopic operations can be used to simulate the effects of external effective fields (e.g. forces), which could allow one to perform transport experiments[21,32] within the photonic lattice over long timescales. Similar operations could be used to design dynamical (density-dependent) gauge fields[34], to imprint effective (mean-field) interactions[19], or to engineer space- and time-dependent losses in the photonic lattice[35].

## Methods

**Fabrication.** The photonic lattices were fabricated inside borosilicate substrates (Corning Eagle[2000]) using ultrafast laser inscription[11]. The substrate was translated at 8 mm/s once through the focus of sub-picosecond laser pulses (350 fs, 500 kHz, 1030 nm) to fabricate each waveguide. The pulse energy of the laser was optimised to realise tightly confined single mode waveguides for a desired wavelength.

For emulating Dirac fermions and the anomalous Floquet topological phase, we used synchronously bent waveguide pairs to turn the bonds on and off; see ref. [6]. Initially (i.e. at $z = 0$), all the waveguides in the lattices are well separated such that the inter-waveguide couplings are insignificant. To turn on coupling between any desired waveguide pair, we reduce the inter-waveguide separation by synchronously bending the waveguide axes. The waveguides then propagate parallel to each other for a certain length and finally separate in a reverse manner. The coupling between two such bent waveguides is equivalent to an effective tight-binding coupling between two straight neighbouring waveguides. The effective bond strength depends on the geometry of the waveguide pair. For the precise control of the bond strength, we tune the wavelength of the excitation light.

**Linear and ring cavity schemes.** The experimental setup for the time-resolved state-recycling is shown in Supplementary Fig. 1. In the experiment, light at 39 MHz pulse repetition rate and a desired wavelength (determined by a bandpass filter, F) is filtered from a broad-band supercontinuum source (NKT Photonics). The beam splitter, $BS_1$, reflects ≈10% of this light which enters the ring cavity (formed by $M_{2-5}$). For precise imaging, the input facet of the lattice is imaged on a CCD camera to observe the lattice sites, input state, and the output state after the

first pass. The optical mode of each waveguide at the output of the photonic lattice is imaged onto individual SPADs of the Megaframe (MF32); [similar devices are now supplied commercially by Photon Force Ltd]. For the experiments shown in Figs. 1c and 2, where the state-recycling is performed using a linear cavity, the aforementioned lattice is replaced by lattices with silver-coated facets (shown in the green-dotted inset) and $BS_3$ is replaced by a mirror to reflect the output state to the MF32. Supplementary Fig. 2a shows an optical micrograph of the MF32 camera consisting of $32 \times 32$ SPAD array.

Time-correlated single-photon counting by the silicon-based SPAD array provides access to both spatial and temporal information. The data processing method for the driven 1D lattice (presented in Fig. 1e–i) is briefly summarised in Supplementary Fig. 2b, c. Supplementary Figure 2b shows the spatial information i.e. intensity distribution summed over four and a half round trips and the normalised temporal information for some specific pixels (indicated by the arrows) are presented in Supplementary Fig. 2c. It should be noted that the temporal separation ($\tau_s$) between two consecutive peaks is determined by the length of the cavity. For this particular experiment, we used a 15-mm-long cavity, hence, $\tau_s \sim$ 150 ps. In this situation, it is expected that the temporal separation $\tau_s$ will be 3 bins (i.e. time-steps), as observed in Supplementary Fig. 2c. Figure 1e–i show the evolution of the intensity distribution along the lattice which was obtained considering the peak intensities of the recorded signals [Supplementary Fig. 2c].

**Intensity and phase recycling in the ring cavity.** To demonstrate that an optical state (both phase and intensity) is recycled in the ring cavity, we perform the following experiment. A symmetric directional coupler, formed by two evanescently coupled identical straight optical waveguides, was fabricated and placed inside the ring cavity. The coupling strengths for this device were measured to be $J = 0.046$ and $0.038$ mm$^{-1}$ at 780 nm and 750 nm wavelengths, respectively. Note that the coupler is an optical analogue of two coupled potential wells supporting two non-degenerate eigenstates with a relative energy offset $2J$. First, we launched optical pulse trains at 780 nm into waveguide-1 and measured output intensities at both waveguides in a time-resolved manner. The blue and red solid lines in Supplementary Fig. 3a shows the expected variation of light intensities, $I_1$ and $I_2$, as a function of the dimensionless parameter, $Jz$. The red and blue squares indicate the measured intensities at 780 nm wavelength for four consecutive round trips. Next, we used pulse trains at 750 nm to reduce the coupling strength and the corresponding intensities are indicated by red and blue circles in Supplementary Fig. 3a. The distributions of light intensity measured after each round trip can only be observed if both the phase an amplitude of the state is preserved during the state-recycling process, confirming this to be the case.

In addition, a time-resolved interference experiment was performed at 750 nm. Supplementary Fig. 3b–e show interference fringes (between the two output modes) detected by the MF32 for four consecutive round trips. The fringes are rotated at 45° because the waveguides were oriented at that angle with respect to the vertical axis. The $\pi$ phase shift observed in Supplementary Fig 3d, e compared to b, c is a well-known characteristic of a directional coupler—after the full transfer

of light, i.e. $Jz > \pi/2$, the relative phase between the optical modes of the waveguides exhibit a phase shift of $\pi$.

## Data availability

Raw experimental data will be made available through Heriot-Watt University PURE research data management system. (https://doi.org/10.17861/490f2eff-e1a8-45d0-846d-3444b42af3f1).

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

## Acknowledgements

This work was funded as part of the UK Quantum Technology Hub for Quantum Communications Technologies—EPSRC grant no. EP/M013472/1, and by the UK Science and Technology Facilities Council (STFC)—STFC grant no. ST/N000625/1. N.G. is financially supported by the FRS-FNRS (Belgium) and the ERC TopoCold Starting Grant. P.Ö. acknowledges support from EPSRC grant no. EP/M024636/1. We thank R.K. Henderson for providing the SPAD array used in this work. We also thank E. Andersson, M. Hartmann, H.M. Price, A. Spracklen and M. Valiente for helpful discussions.

## Author contributions

R.R.T. conceived the state-recycling technique and the enabling experimental methods. S.M. designed and fabricated the photonic devices, conducted the state-recycling experiments with H.K.C., and analysed the data with inputs from H.K.C. and R.R.T. S.M. and N.G. performed all theoretical calculations. S.M. and N.G. wrote the manuscript, which was finalised by the team. S.M., H.K.C., R.R.T., N.G. and P.Ö. discussed the results. R.R.T. and N.G. supervised the experimental and theoretical aspects, respectively.

## Additional information

**Competing interests:** The authors declare no competing interests.

