## [Peer Review File · Nature Communications]

Reviewers' comments:

Reviewer #1 (Remarks to the Author):

The paper is suitable for publication in Nature Communications as is.

Reviewer #2 (Remarks to the Author):

The revised manuscript is significantly improved. In particular, the authors have demonstrated good performance of the linear-cavity recycling method for bandstructures with non-trivial dispersion.

In the linear-cavity scheme, it was found that the recycling does not match initial expectations due to small angular deviations at the facets, but the authors argue that this can instead be thought of as the effect of a periodic driving electric field. This is an interesting idea, and this set of results that may stimulate progress in the field.

However, it should be explained how the parameters of the simulations were chosen/fitted. I also suggest finding a better way to plot the results in Fig. 2e-h, if possible; currently, it is difficult to make out the quality of the agreement between "experiment" and "theory (angled facets)" for subplots g-h without zooming into the figure and studying it very closely.

As for the ring-cavity scheme, I am afraid the situation is still murky. In their response, the authors state that they used a waveguide coupler to show that the input re-injected into their anomalous Floquet lattice is phase coherent. But that was not really the point. The issue is that in this particular anomalous Floquet lattice, if light is injected into a single site, after one evolution cycle it remains concentrated (ideally) on a single site. The relative phase between different sites is irrelevant, since the magnitude is zero on all the other sites. But for any lattice that is more complicated, the wavefunction after evolution by one cycle will not be concentrated onto one site. The anomalous Floquet lattice used in the present paper is a special dispersionless case; for more general anomalous Floquet lattices, one observes diffraction, as shown in Nature Physics 13, 611 (2017). In the more general case, can the light be re-injected including full phase information? Or is this method limited to the special lattices with input/output on a single site? After reading the manuscript, SM, and response letter several times, this is still unclear to me. Even if the answer is no, the authors can at least provide a clear explanation of the issue and the limitation of the ring-cavity scheme.

In my view, if the above issues are sorted out, the manuscript is suitable for Nature Communications.

Reviewer-1 (Remarks to the Author):

Comment– The paper is suitable for publication in Nature Communications as is.

Reply – We thank the referee for this comment.

Reviewer-2 (Remarks to the Author):

Comment – The revised manuscript is significantly improved. In particular, the authors have demonstrated good performance of the linear-cavity recycling method for band structures with non-trivial dispersion.

In the linear-cavity scheme, it was found that the recycling does not match initial expectations due to small angular deviations at the facets, but the authors argue that this can instead be thought of as the effect of a periodic driving electric field. This is an interesting idea, and this set of results that may stimulate progress in the field.

However, it should be explained how the parameters of the simulations were chosen/fitted. I also suggest finding a better way to plot the results in Fig. 2e-h, if possible; currently, it is difficult to make out the quality of the agreement between "experiment" and "theory (angled facets)" for subplots g-h without zooming into the figure and studying it very closely.

Reply – We thank the referee for this comment. We have now mentioned all the parameters associated with Fig. 2. In the experiment, we measured the tunnelling strength and the facet angles [now mentioned in the main text, page 9]. In the numerical calculation, the required parameters are the tunnelling strength and the inter-waveguide phase shifts due to the facet angles [mentioned in the supplementary page 16].

We considered presenting the information in Fig 2E-H in a different way – namely by plotting the inverse participation ratio which is a measure of localisation/delocalisation. However, we do think the current presentation of these figures shows the maximum possible information. In response to the Referee’s comment, we have therefore improved and magnified Fig. 2E-H such that a reader can see the agreement between experiment and numerical results without zooming in on the figure.

Comment – As for the ring-cavity scheme, I am afraid the situation is still murky. In their response, the authors state that they used a waveguide coupler to show that the input re-injected into their anomalous Floquet lattice is phase coherent. But that was not really the point. The issue is that in this particular anomalous Floquet lattice, if light is injected into a single site, after one evolution cycle it remains concentrated (ideally) on a single site. The relative phase between different sites is irrelevant, since the magnitude is zero on all the other sites. But for any lattice that is more complicated, the wavefunction after evolution by one cycle will not be concentrated onto one site. The anomalous Floquet lattice used in the present paper is a special dispersionless case; for more general anomalous Floquet lattices, one observes diffraction, as shown in Nature Physics 13, 611 (2017). In the more general case, can the light be re-injected including full phase information? Or is this method limited to the special lattices with input/output on a single site? After reading the manuscript, SM, and response letter several times, this is still unclear to me. Even if the answer is no, the authors can at least provide a clear explanation of the issue and the limitation of the ring-cavity scheme.

In my view, if the above issues are sorted out, the manuscript is suitable for Nature Communications.

Reply – First of all, we would like to stress that both phase and intensity can be recycled in our ring-cavity scheme for any desired photonic lattice – our method is *not limited to special lattices with input/output on a single site*. We now realise that the explanation in the method section and Supplementary material was not clear. In particular, the Supplementary Figure 3 shows a completely separate set of experimental results which are not associated with anomalous Floquet lattice. In other words, Supplementary Figure 3 does not show the relative phase of the anomalous Floquet lattice. We would like to further elaborate these aspects to the referee.

The Referee correctly mentioned that light remains localised (ideally) on a single site for the anomalous Floquet lattice. This is why, the relative phase of that edge state after different round trips is not relevant. Hence, to demonstrate phase recycling and obtain phase information it is required to use a photonic device with at least two optical modes (that can interfere) at the output.

To demonstrate that an optical state can be recycled in our ring-cavity with phase and intensity preserved, we used a *two-site lattice*, as motivated below. As discussed in the method section (page 17) and Supplementary Figure 3, we demonstrated the robustness of the recycling by launching light into one waveguide of a two-waveguide coupler (a directional coupler) and then by measuring phase and intensity in a time-resolved manner. A symmetric directional coupler is an optical analogue of two coupled potential wells supporting two non-degenerate eigenstates with a relative energy offset $2J$. The electric field envelope at the two waveguides is governed by the following coupled mode equations: $i\partial_z\psi_{1,2} = -J\psi_{2,1}$. The above-mentioned input state, $\psi_1(z=0)=1$, equally overlaps with both eigenstates ($\psi_1 \pm \psi_2$) and the evolution of the optical fields (the analogous wavefunction) exhibit a well-known dynamics. The optical intensity evolves continuously and the relative optical phase exhibits discrete jumps along the propagation distance (the analogous time).

Reply Figure 1: Simplified sketch showing the state-recycling scheme in a ring-cavity – the input state (gray) and the first pass (dashed red) are shown only.

As shown in Reply Figure 1, the evolution of an optical state can be described by the evolution operators \hat{U}_{device} (determined by the analogous Hamiltonian of the device, in this case a directional coupler) and $\hat{U}_{\text{free space}}$ (state recycling outside the photonic device). In our experiments, $\hat{U}_{\text{free space}}$ is unitary (provided that we renormalise the intensity in order to account for global losses). In Supplementary Figure 3, we show that the evolution of the optical state (phase and intensity) inside the directional coupler [determined by \hat{U}_{device}] agrees with the theory up to four different round trips which would not occur if $\hat{U}_{\text{free space}}$ were not relative phase and intensity preserving.

In response to the Referee’s comment, we have now carefully highlighted the following aspects.

(1) Main text, page 10 paragraph 2: We now present more details on the experiments demonstrating intensity and phase recycling where we also stress that this was a separate experiment demonstrating that light can be re-injected with relative phase and intensity preserved in a general situation.

(2) In the method section (page 17, paragraph 2, line 5): we now mention that a coupler supports two *non-degenerate energy states* – to further clarify to a general reader that an initial state overlapping with these eigenstates will not behave like a dispersionless state.

REVIEWERS' COMMENTS:

Reviewer #2 (Remarks to the Author):

I am fully satisfied with the latest response and revisions from the authors, and recommend the manuscript for publication in Nature Comms.